# A Current Review on the Role of Prebiotics in Colorectal Cancer

**Anna Shrifteylik** [1,†], **Morgan Maiolini** [2,†], **Matthew Dufault** [3,†], **Daniel L. Austin** [4,†], **Bobban Subhadra** [5], **Purushottam Lamichhane** [6,*] and **Rahul R. Deshmukh** [7,*]

1. School of Pharmacy, Lake Erie College of Osteopathic Medicine, Bradenton, FL 34211, USA; ashriftey102269@rx.lecom.edu
2. CVS Health, Philadelphia, PA 17601, USA; mmaiolini21274@rx.lecom.edu
3. Arizona College of Osteopathic Medicine, Midwestern University, Glendale, AZ 85308, USA; matthew.dufault@midwestern.edu
4. School of Pharmacy, Lake Erie College of Osteopathic Medicine, Erie, PA 16509, USA; daustin@lecom.edu
5. BIOM Pharmaceutical Corporation, Sarasota, FL 34234, USA; bbobban@gmail.com
6. Novocure, Wayne, PA 19087, USA
7. College of Pharmacy, Midwestern University, Glendale, AZ 85308, USA
* Correspondence: lamichhanepurru@gmail.com (P.L.); rdeshm@midwestern.edu or rahulrdeshmukh@hotmail.com (R.R.D.)
† These authors contributed equally to this work.

**Abstract:** Colorectal cancer (CRC) is one of the leading causes of death in the United States and worldwide. Recent evidence has corroborated a strong correlation between poor diet and the development of CRC, and further research is being conducted to investigate the association between intestinal microbiome and the development of cancer. New studies have established links with certain foods and synthetic food compounds that may be effective in reducing the risk for carcinogenesis by providing protection against cancer cell proliferation and antagonizing oncogenic pathways. Prebiotics are gaining popularity as studies have demonstrated chemo-preventive as well as anticancer potential of prebiotics. This paper aims to discuss the wide definition and scope of prebiotics by reviewing the studies that provide insights into their effects on human health in the context of colorectal cancer.

**Keywords:** prebiotics; probiotics; synbiotics; dietary fiber; colorectal cancer; gut microbiota





## 1. Introduction: Definition of Prebiotics and Its Evolution

In June of 1995, a paper published in the *Journal of Nutrition* first introduced the concept of prebiotics to the scientific community. First described by Glenn Gibson and Marcel Roberfroid, the original definition of prebiotics included "non-digestible food ingredient that beneficially affects the host by selectively stimulating the growth and/or activity of one or a limited number of bacteria in the colon, and thus improves host health" [1]. Since then, the definition of prebiotics has been continuously evolving with no clear consensus between regulators like the U.S. Food and Drug Administration (FDA) and dietary product manufacturers. A meeting held in 2007 by the Food and Agriculture Organization of the United Nations described prebiotics as 'a nonviable food component that confers a health benefit on the host associated with modulation of the microbiota' [2]. Additionally, the meeting's published report acknowledged that the industry lacked consistent guidelines governing the usage of the term 'prebiotic'. In 2007, the prebiotic market offered over 400 prebiotic food products with more than 20 companies producing oligosaccharides and fibers which they labeled as prebiotics [2]. Today, the number of prebiotic-containing foods and beverages exceeds 1300 [3].

Additional organizations have proposed definitions for prebiotics, with the most recent definition being from the International Scientific Association for Probiotics and Prebiotics which states that prebiotic is considered as "a substrate that is selectively utilized by host microorganisms conferring a health benefit" [4]. According to the World Gastroenterology Organization (WGO), probiotics are defined as live microorganisms which can be

formulated into different products, including foods, drugs, and dietary supplements. These microorganisms help maintain and improve the body's normal microflora [5–7]. With the concept of prebiotics continuously evolving, it is important for the scientific community to continue developing the definition as new research is made available.

Amidst the everchanging scientific landscape surrounding prebiotics today, the fundamental criteria proposed by Roberfroid are still widely accepted. These include additional criteria to his original definition of prebiotics. A food ingredient qualifies as a prebiotic if it is (1) neither hydrolyzed nor absorbed in the upper part of the gastrointestinal tract; (2) a selective substance for one or a limited number of beneficial bacteria commensal to the colon, which are stimulated to grow, are metabolically activated, or both; (3) able to alter the colonic flora in favor of a healthier composition; and (4) able to induce luminal or systemic effects that are beneficial to the host health. Consequently, Roberfroid described fructo-oligosaccharides (FOS), inulin, and galacto-oligosaccharides (GOS) as the only "true" prebiotics, which meet these qualifications [8]. While no stable definition currently exists, a prebiotic must only meet three criteria: (1) resistant to hydrolysis and absorption, (2) selective growth or activity in the intestines that is associated with positive health outcomes, and (3) fermented by the intestinal flora [9]. The first two criteria were proposed by Roberfroid and the third adds the additional requirement of fermentation.

As for the general consumer, the definitions of prebiotics and dietary fibers can be difficult to delineate. Various health organizations continue to refine these definitions. In June of 2018, the FDA identified eight additional non-digestible carbohydrates ("mixed plant cell wall fibers; arabinoxylan; alginate; inulin and inulin-type fructans; high amylose starch (resistant starch 2); galacto-oligosaccharide; polydextrose; and resistant maltodextrin/dextrin") that meet the definition of dietary fiber of Nutritional Facts labels [10]. Of these fibers, inulin and inulin-type fructans along with GOS are widely accepted as prebiotics by the scientific community; however, further research is needed to investigate the effects of the additional fibers. A recently published review by Mysonhimer and colleagues provided an updated summary of clinical trials that observed tolerance and side effects of these non-digestible carbohydrates. While it is well known that inulin and GOS are generally well tolerated up to 5 g and 20 g, respectively, this review provides dose recommendations for the additional less studied fibers. The mixed cell wall fibers β-glucans have not been demonstrated to be well tolerated at doses as low as 3 g, whereas pectin appears to be well tolerated up to 36 g. Arabinoxylan (up to 15.1 g), alginate (up to 3.75 g), polydextrose (up to 12 g), and resistant maltodextrin/dextrin (up to 12 g) are all generally well tolerated. Higher doses frequently result in side effects of flatulence and diarrhea, but doses within the appropriate ranges help maintain healthy stool viscosity and bowel movement frequency. More studies are needed to determine the effect that these fibers have on the microbiome composition and gut health to determine their status as prebiotics [11]. This information would provide individuals with the knowledge they need to safely incorporate these fibers into their diets.

## 1.1. Prevalence of Prebiotics and Rationale behind Usage

As the importance of gut microbiota diversity is receiving recognition in the health community, the consumption of prebiotics and probiotics has gained traction. Though prebiotics are a newer scientific concept than probiotics, they are quickly achieving popularity in use. According to the National Health Interview Survey (NHIS), the use of prebiotics or probiotics was four times higher in the year of 2012 than it was in 2007 (1.6% and 0.4%, respectively, of all U.S. adults), becoming the third most commonly used non-vitamin, non-mineral dietary supplement [12]. Recent prebiotic global market trends suggest that the compound annual growth rate of the industry will be approximately 14.9% from 2022 to 2030; the most notable drivers of this expansion are enhanced consumer awareness of prebiotic health benefits leading to increased consumption and increased utilization of prebiotic fibers in the food industry (particularly as sugar substitutes) [13].

Host benefits from prebiotic consumption are largely related to the resulting modulation of microbial populations within the gut microbiome. The gut flora consists of a diverse population of different microbial organisms which play a role in many functions of the human body ranging from metabolism to immunity and infection prevention. Probiotic bacteria generate short-chain fatty acids (SCFAs) used for host energy production, and they also help to generate vitamin K, folic acid, and amino acids (arginine and glutamine). The bacteria can also aid in certain drug metabolisms. Their immunologic effects are still being researched; however, they have been shown to stimulate the production of immunoglobulin A, anti-inflammatory cytokines, and regulatory T cells [14,15]. These immunologic effects have been seen in rodent studies where investigators researched the effects on tumor growth and the expression of immunologic cells and factors [15]. Lastly, the composition of the gut flora directly influences the growth potential of pathogenic bacteria; probiotic microbes compete with pathogens and certain strains can eliminate pathogenic bacteria via phagocytosis [14,16]. Thus, it is important to keep the microbiota at homeostasis.

When prebiotics are consumed along with probiotics, they act as an energy source for the live microorganisms in the gut and facilitate their proliferation. The combination of prebiotics and probiotics is termed synbiotics [17], and the latest definition of synbiotics is described as "a mixture comprising live microorganisms and substrate(s) selectively utilized by host microorganisms that confers a health benefit on the host" [18]. This combination promotes the growth of existing strains of beneficial microorganisms in the colon, and improves the survival, implantation, and growth of newly added probiotic strains [19]. Synbiotics can be further categorized as "synergistic" where the combination is aimed at promoting the growth of the newly introduced probiotics, or "complementary" where the prebiotic can promote the growth of resident probiotic species as well as the newly introduced probiotics [18]. In vivo studies of synbiotics have demonstrated their potential anticarcinogenic and antimutagenic properties [19]. These benefits attributed to prebiotics are related specifically to the stimulation of beneficial bacterial, production of SCFAs, modulation of gene expression, upregulation of nutrient absorption, alteration of metabolism, and alteration of immune response [20].

In a recently completed study, the consumption of date palm fruits was observed in human subjects. These fruits are plentiful in oligosaccharides and polyphenols. The investigators looked at healthy subjects and the changes in the intestinal microbiota. Although there was a marked increase in *Bifidobacteria* when fecal samples were analyzed, the change was not statistically significant. The authors also discussed that polyphenols extracted from the fruits appeared to inhibit the proliferation of Caco-2 colorectal adenocarcinoma cells in vitro, suggesting anti-carcinogenic activity [21]. These whole food sources of natural prebiotic compounds should be considered in the prevention of colorectal cancer due to their nutrition and potential anticancer properties. In addition, these whole food sources are typically more easily accessible when compared to other supplements or drugs.

Lastly, it should be noted that side effects are possible as previously mentioned, and most problems are seen in the gastrointestinal tract. Side effects include cramping, nausea, flatulence, and bloating due to water osmosis into the intestinal lumen and the production of $CO_2$ gas in the fermentation of prebiotics. Some patients have also reported headaches [11,22,23]. It is important to note that supplements as a category are regulated under the Dietary Supplement Health and Education Act (DSHEA) by the Food and Drug Administration (FDA) [24]. However, the FDA does not review the safety and efficacy of each individual supplement before it is marketed [24].

### 1.2. Colorectal Cancer: Epidemiology, Potential Causes, and Key Players

The American Cancer Society has named colorectal cancer as the third leading cause of cancer-related death in the United States. Worldwide, CRC is the third most common type of cancer and the fourth cause of cancer-related deaths [25]. While early screening and improvements in treatment have attributed to a decrease in death rates, the projected number of deaths due to colon and rectal cancer in 2021 was estimated to be roughly

52,980 men and women in the United States [26]. Currently, screening is recommended after the age of 50 or earlier if the patient has inflammatory bowel disease, Lynch Syndrome, or a family history of CRC [27]. Current trends show that overall CRC incidence has been declining for the past 40 years. However, this decline has started to reverse as younger individuals (less than 55 years old) have seen a 9% increase in incidence over the last 25-year period [28]. Although the exact cause of this trend is uncertain, physicians and scientists postulate that processed foods, obesity, and toxins from alcohol and tobacco likely play a role. A growing body of research also suggests that dairy and beef products may contain viral DNA particles that may play a role in oncogenesis [29]. However, more research is needed to gain insight into the various mechanisms that predispose individuals to CRC. With this changing trend also comes a need to reevaluate CRC screening criteria. Fortunately, colonoscopies provide patient screening with the added benefit of polyp removal and sampling capabilities. Once a polyp is excised during a colonoscopy procedure it can be evaluated for signs of pathology, and for these reasons it is considered the most thorough and effective CRC screening tool. Other tests include blood tests (liquid biopsy) which screens for *SEPT9 gene mutations*, and fecal samples to detect blood in the stood which may potentially be a sign of CRC or other abnormalities such as hemorrhoids; however, these tests are not a substitute for the imaging, screening, and tissue retrieval capabilities of a colonoscopy [30].

In normal human physiology, cell division and proliferation are tightly controlled processes with many checks and balances. These checkpoints throughout the cell cycle ensure genomic integrity and controlled replication [31]. Cancer develops in the body when abnormal cells escape the body's intrinsic regulatory systems enabling the cells to proliferate uncontrollably while spreading into surrounding tissues [32]. Specifically, in CRC, the polyps develop in the inner lining of the colon or rectum. Adenomatous polyps, which contribute to roughly 96% of all colorectal cancers, are considered non-malignant when detected. Alternatively, hyperplastic and inflammatory polyps are the most commonly found polyps which typically remain benign [33].

The walls of the colon consist of an inner mucosal layer, a middle submucosal layer, and an external muscular layer. Polyps originate in the innermost mucosal layer. The abnormal cells within the polyp may become cancerous and grow outwards into subsequent layers until they finally access blood vessels and lymph nodes [32]. This progression of the disease is clinically categorized by stages determined by the American Joint Committee on Cancer (AJCC) TNM system. The TNM Staging System is based on the size and extent of the tumor (T), the extent of spread to the lymph nodes (N), and the presence of metastasis (M) [34].

Several gene dysregulation events have been implicated in the carcinogenesis and progression of CRC. Several studies have examined the influence of prebiotics on the expression of key genes that regulate programmed cell death (apoptosis) such as CRC, BCL-2, Bax, Caspases, and Survivin. The BCL-2 gene helps regulate cell survival and is considered an oncogene that can result in decreased cell apoptosis. Bax is an additional gene with implications in cell-mediated death, but Bax can be inhibited by BCL-2. These two genes are responsible for cell death at the mitochondrial level, but other factors such as Caspase 3, Caspase 9, and Survivin can mediate the process throughout the cell. Caspases, which are cysteine proteases, can either initiate or continue the internal process of apoptosis. Survivin is an apoptosis inhibitor that inhibits the caspases [35]. One major regulator of cell cycle progression, p53, suppresses tumor formation by aiding in the regulation of cell death and cell arrest when the genome is unstable at the $G_1$ and $G_2$ checkpoints [36]. Figure 1 exhibits the common expression trends of these genes in cancer cells. When looking at options to target these genes and their products, the therapeutic goals are to enhance cell death and decrease the inhibitors of apoptosis in cancer cells.

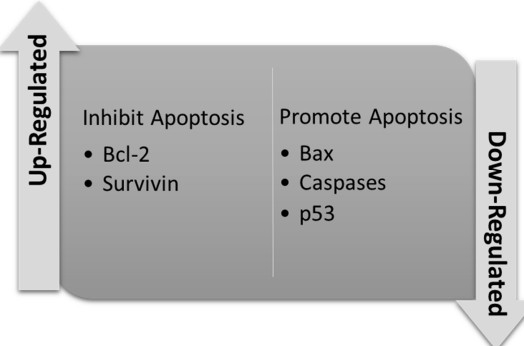

**Figure 1.** Key genes and their expression in cancer. Note: the goal of treatment is to enhance apoptosis in cancer cells.

There are various risk factors that can increase a person's predisposition to colorectal cancer. Modifiable factors include inactivity, smoking, and heavy alcohol use. Factors such as these can be avoided or mitigated by diet, exercise, and minimizing the use of alcohol and smoking [37,38]. Nonmodifiable risk factors such as race, family history, socioeconomic status, and some conditions such as inflammatory bowel disease, Lynch Syndrome exist and increase a person's risk for carcinogenesis [37].

The American Institute for Cancer Research (AICR) recommends several lifestyle choices which are linked to a lower incidence of cancer. Consistent exercise and the consumption of healthier foods are behaviors that have been extensively shown to help prevent cancer. AICR recommends a diet that includes a variety of fruit, vegetables, whole grains, and legumes, such as beans. This diet offers a variety of options that inherently increases the percentage of fiber consumed [39]. In the following section, various classes of prebiotic fibers are discussed as well as their understood mechanisms and specific whole food sources. This information is presented with the goal of providing a knowledge base for the selection of prebiotics to include in a healthy diet that may help to prevent CRC.

### 1.3. Current Classification of Prebiotics and Mechanisms Regarding CRC

Currently, the universally accepted and most heavily studied prebiotics are inulin, FOS, oligofructose, and GOS. These prebiotics are neither absorbed nor digested in the small intestines and are classified as fiber. Inulin, FOS, and oligofructose are categorized as fructans while GOS is categorized under galactans [40]. More recently, potential new varieties of prebiotics have been demonstrated to promote the growth of beneficial bacteria such as protein-oligosaccharide conjugates, human milk oligosaccharides (HMOs), and non-carbohydrate molecules including polyphenols, fats, and various plants and herbs [41,42]. Here, we will discuss the mechanisms and whole food or supplemental sources of fructans, galactans, and promising prebiotic candidates so that their potential implications surrounding colorectal cancer may be better understood (see Table 1 as a summary and reference).

#### 1.3.1. Fructans and Galactans

Fructans are oligosaccharides that consist primarily of fructose and fructosyl units. These sugars are polymerized via beta linkages that make them resistant to degradation via human digestive enzymes. The fructans group includes the prebiotics inulin, FOS, and oligofructose. Inulin is the longest of these polysaccharides with a degree of polymerization (DP) between 3 and 60 fructose monomers followed by oligofructose (DP of 2–20 fructose monomers) and FOS (DP of 3–9 fructose monomers linked to sucrose). These prebiotics travel in the gastrointestinal (GI) tract until they reach the colon where they are fermented. They have been widely studied in different populations, children to adults, for a variety of outcomes, such as constipation, weight loss, appetite suppression, and respiratory infection protection [43,44]. It has been shown that fructan prebiotics are primarily fermented by

*Bifidobacteria* in the colon. These microbes express efficient importers for these molecules and enzymes that cleave the beta linkages to liberate fructose monomers. *Bifidobacteria* then produce SCFAs as byproducts of this metabolic process [40]. The growth of *Bifidobacteria* in the human gut following inulin supplementation has also been confirmed in a recent double-blinded controlled trial [45].

Galactans are oligosaccharides derived from galactose and glucose monomers, and the GOS prebiotics fall under this category. GOS may be further designated as alpha-GOS or beta-GOS depending on the sugar linkages. Alpha-GOS is naturally occurring in foods, whereas beta-GOS is produced synthetically. GOS, like the fructans prebiotics, resist digestion in the human GI tract until they encounter microbes in the colon that facilitate their catabolism via fermentation. *Bifidobacteria* are the strongest consumers of galactans prebiotics such as GOS and constitute a greater fraction of the gut microbiome following the consumption of GOS in human subjects. *Lactobacillus* numbers have also been observed to increase after GOS supplementation [9,40].

Both fructans and galactans prebiotics can be found in isolated supplement form or naturally in whole foods. Currently, there are several over the counter formulations available on the market as a combination of prebiotics with probiotics as well as products composed of solely prebiotics. When prebiotics are listed on the Supplemental Facts label, they are either listed as 'Dietary Fiber' or as specific components. These specific ingredients such as chicory root, dandelion greens, Jerusalem artichoke, garlic, onions, leeks, asparagus, bananas, burdock root, yacon root, and jicama root, are listed as prebiotic ingredients based on their inulin/FOS content [43]. Alpha-GOS can be found naturally in legumes and various grains. Beta-GOS is synthesized in an enzymatic reaction utilizing beta-galactosidase and lactose; it may be added to various foods or used in isolation as a supplement [46].

It is important to note that whole food sources of prebiotics are the most accessible and provide a wide range of nutritional benefits; however, they are also the most variable in terms of dosage. In contrast, supplement dosing can be tracked and modified appropriately to achieve a desired outcome [47]. Prebiotic dosing via supplements may also help consumers to accurately gauge when the beneficial bifidogenic effects are stimulated; for FOS this is 10 g/day, 2.5–5 g/day for inulin, and 7 g/day for beta-GOS. At these doses, *Bifidobacteria* and subsequent SCFA concentrations are significantly increased in the colon; however, when doses for any of these prebiotics reach 40–50 g/day, complications with osmotic diarrhea and nausea have been noted as these compounds draw water into the intestinal lumen. Additionally, at bifidogenic doses of prebiotics, side effects of flatulence and borborygmi are common due to the microbial production of carbon dioxide, methane, hydrogen, and nitrogen gases [23]. Ultimately, the consumer decides their optimal use of prebiotics, and it is critical that more research is conducted to determine the effects of various routes of prebiotic consumption on long-term health outcomes.

### 1.3.2. HMOs

Human milk oligosaccharides (HMOs) are a group of over 200 carbohydrate compounds that are highly abundant in human milk and are being considered as a new class of prebiotics [48]. HMOs are minimally digested in the gastrointestinal tract and reach the colon intact, where they shape the gut microbiota [49]. HMOs are especially abundant in the early colostrum secretions from the human mammary gland at 20–25 g/L and subside to 10–15 g/L in following milk secretions [50,51]. HMOs have been proven to stimulate gut adaptation and reduce the incidence of necrotizing enterocolitis (NEC) in infants [52–55]. Recently, HMOs have been shown to coat various microbial pathogens, thereby blocking their ability to bind to gut epithelial cells which helps to prevent infection, inflammation, and colonization [54]. There are three major HMO categories: fucosylated neutral HMOs, sialylated acidic HMOs, and non-fucosylated neutral HMOs. Technologies and methods to synthetically reproduce these HMOs open avenues for industrial employment of HMOs in various foods and supplements [56]. Many companies are developing branded HMOs

for the dietary supplement sector (e.g., CARE4U brand is fucolyated HMO developed by Dupont-Danisco for precision nutrition). As prebiotic nutrition is a major component in shaping the gut microbiome, specific prebiotic fibers such as HMOs may be a critical component in cancer prevention or treatments. In vitro studies have shown that HMOs have anti-proliferative and growth arrest effects; CRC cell lines treated with HMOs have increased expression of cyclin D1 and become arrested at the G2/M checkpoint [57,58]. The proposed mechanism is that HMOs directly interact with epidermal growth factor receptors in the cultured CRC cells to activate ERK1/2-mediated cyclin B1 expression and promote the p53/p38/p21 cascade to arrest the cells at the G2/M checkpoint [57]. HMOs also promote *Bifidobacteria (B. infantis)* and *Lactobacillus* growth which metabolize HMOs and produce SCFAs [59]. Recently, a study was performed to engraft *B. infantis* in adult subjects through a synbiotics approach with simultaneous HMO supplementation. The investigators found that at doses of 18 g/day *B. infantis* became abundant in most subjects' microbiomes. Fecal butyrate levels were elevated in the human subjects, and in murine models enteropathogens were outcompeted by *B. infantis* [60]. This study demonstrates the efficacy of HMO supplementation in adults; it also prompts further investigation to determine if the benefits that HMOs confer to infants can also be seen in adults and if HMOs demonstrate anti-neoplastic activity in vivo. Additionally, HMOs have been implicated in immunomodulatory effects which will be further discussed in this section.

### 1.3.3. Protein–Oligosaccharides

One emerging class of prebiotics is protein-conjugated oligosaccharides. These prebiotics are developed by the covalent linkage of various proteins to prebiotic fibers via the Maillard reaction which links reducing sugar molecules to amino acids. This covalent modification can be used to increase the bioavailability of proteins as an energy source to beneficial probiotic strains. Increased protein delivery to prebiotic strains is achieved through the protein–oligosaccharide conjugate resisting digestion by GI proteases, thereby promoting the growth of probiotic strains capable of importing and metabolizing the prebiotic [61]. In a recent study, Seifert et al. synthesized GOS molecules bound to lactoferrin hydrolysate and reported that this conjugate resisted simulated human GI digestion and resulted in a growth rate of *Lacticaseibacillus casei* two times that of the nonconjugated GOS and lactoferrin in vitro. Lactoferrin was selected for its high cysteine content which has been demonstrated to enhance growth of *Lactobacilli* and *Bifidobacteria* in vitro. In this study, lactoferrin was pretreated with pepsin before conjugation to ensure that the product peptides would not undergo further cleavage in the small intestine by human endogenous pepsin. In a separate study, Zhong et al. prepared soy protein isolate conjugated to isomaltooligosaccharides, xylooligosaccharides, and GOS. When these conjugates were encapsulated with *Lacticaseibacillus casei* as a supplement, a significant increase in *L. casei* was observed with a simultaneous decrease in pathogenic bacteria under simulated digestive conditions [41]. These protein–oligosaccharide compounds demonstrate a previously unexplored targeted approach to prebiotic use and have implications for the selective modulation of the human microbiome for therapeutic purposes, especially when employed as synbiotics. However, there is a need for in vivo experimentation in preclinical animal models to determine the efficacy of these compounds in the heterogenous environment of the mammalian gut.

### 1.3.4. Plant Polyphenols

While not currently classified as prebiotics, plant polyphenols have received increasing interest for their effects on the host microbiome and gut health. Polyphenols are metabolites of plants consisting of hydroxylated aromatic rings that resist degradation and absorption in the human GI tract [62]. Over 90% of ingested polyphenols remain unabsorbed as they reach the colon where they can be processed by resident microbes [40]. While the precise mechanisms by which polyphenols stimulate SCFA production via probiotic bacteria remain to be elucidated, several studies have observed increased SCFA production

in animal models treated with polyphenol compounds [62]. Furthermore, several studies have reported mechanisms by which polyphenols can modulate the gut endothelial and microbiome composition and exert antioxidant and anticancer effects. Liu and colleagues isolated the polyphenol hydroxysafflor yellow A (HSYA) from the safflour plant (traditionally used in Chinese herbal medicines) and demonstrated that mice fed a high fat diet over the course of 6 weeks while receiving the HSYA supplement experienced protection from lipid-induced dysbiosis. They observed that the intestinal barrier in mice receiving HSYA was strengthened via enhanced expression of KLF4 and Muc-2 (goblet cell marker and mucin, respectively), and increased expression of ZO-1 and occludin (components of the gut epithelial tight junctions). These gut endothelial changes suggested decreased permeability of the gut endothelial layer for protection against infiltration of gut pathogens capable of inducing inflammation and tissue damage. Additionally, increases in SCFA producing bacteria *Butyricimonas* and *Alloprevotella*, and immunogenic *Akkermansia* and *Romboutsia* were observed [63].

In a recent preclinical study, Messaoudene and colleagues identified castalagin, a polyphenol found in camu-camu (CC) berries, as an enhancer of anti-PD-1 immunotherapy in subcutaneous mouse models of sarcoma and breast cancer. Treatment mice received CC extract with anti-PD-1 antibodies and their tumors and spleens were harvested at day 9 of treatment for sarcoma-bearing mice and day 11 of treatment for breast tumor-bearing mice. Mice that received the combination of CC extract with anti-PD-1 antibodies exhibited significant reduction in tumor volumes compared to controls, and there was a high correlation between mice that received this treatment and active CD8+ cytotoxic T cells in the tumors and spleens. The investigators further determined that the anti-tumor effects were specifically attributed to the polyphenol castalagin, and that reduction in tumor size was dependent on the castalagin-induced increase in *Ruminococcaceae*, *Alistipes*, and *Ruthenibacterium lactatiformans* bacteria. Furthermore, castalagin treatment increased the concentration of taurine-conjugated bile acids in the colon which are known to prevent adenoma of the colon [64]. This study provides evidence of a polyphenol directly stimulating growth of colonic microbial species, and this modulation of the gut microbiome directly impacting the efficacy of anti-PD-1 immunotherapy to reduce tumor size. These findings warrant further investigation in preclinical models of additional tumor types such as CRC and in clinical trials with the goal of combating resistance to anti-PD-1 therapy.

### 1.3.5. SCFAs

The health benefits of prebiotics is strongly attributed to their fermentation by intestinal microbes into short-chain fatty acids (SCFAs), which have a wide range of implications for gut flora and the host [9]. SCFAs are fatty acids with less than six carbons, such as acetate, butyrate, and propionate. These carbon-containing molecules exert pleotropic effects on host metabolism and immunity as well as pathogenicity of various disease-causing microbes [65]. Regarding the protective effects against CRC, SCFAs decrease the pH of the intestinal lumen, decreasing the viability of many enteropathogenic species while increasing the proliferation of beneficial *Lactobacilli* and *Bifidobacteria*. This hindrance toward pathogen colonization of the gut can decrease the prevalence of acute and chronic inflammation that could lead to DNA damage and oncogenesis [66]. Recent literature has examined the unique responses of pathogenic microbes to the presence of SCFAs. Some *E. coli* variants may upregulate virulence factors, flagella, or invasive mechanisms to escape high luminal butyrate and propionate concentrations; these resistance mechanisms are capable of inducing the host inflammatory response which can result in tissue damage. In contrast, species such as *Shigella* and strains such as *Salmonella Typherium* and *Campylobacter jejuni* have reduced expression of genes that promote invasion into host epithelium when exposed to SCFAs which protects the host from inflammation [67]. SCFAs have also been implicated in the direct modulation of anti-inflammatory mechanisms, and a growing body of evidence is elucidating the interplay between SCFA production and the host immune system.

**Table 1.** Sources, mechanisms, and effective doses of various classes of prebiotics.

| Prebiotic Classes and Candidates | Whole Food Sources | Key Associated Probiotic Species | SCFAs Produced | Mechanisms of Host Benefits | Minimum Dose to Achieve Benefits | References |
|---|---|---|---|---|---|---|
| Fructans | Chicory root, dandelion greens, Jerusalem artichoke, garlic, onions, leeks, asparagus, bananas, burdock root, yacon root, and jicama root | • *Bifidobacteria* | Yes | • Bifidogenic benefits<br>• Improved immunity and gut epithelial health | 10 g/day | [23,43] |
| Galactans | Beta GOS: synthetic supplement only<br>Alpha GOS: legumes including fava beans, fenugreek, chickpea, and lentils, and raw seeds | • *Bifidobacteria*<br>• *Lactobacilli* | Yes | • Bifidogenic benefits<br>• Improved immunity and gut epithelial health<br>• Lactobacilli induced pH reduction inhibits enteropathogen grown | 5.5 g/day | [23,68] |
| Human milk oligosaccharides | Human breast milk | • *Bifidobacteria (B. infantis)*<br>• *Lactobacilli* | Yes | • Promote gut barrier health and immune tolerance<br>• Decrease prevalence of necrotizing enterocolitis (NEC) in infants<br>• Inhibit progression through G2/M checkpoint in vitro models of CRC | 18 g/day for most consumers | [54,57,59,60,69] |
| Protein–oligosaccharide conjugates | N/A | • *Bifidobacteria*<br>• *Lactobacilli* | Yes | • Prebiotic benefits achieved with potentially smaller doses<br>• Fewer dose-related side effects<br>• Potential for selective probiotic targeting | No in vivo studies to date | [41,61] |
| Plant Polyphenols | Berries, herbs such as cloves, peppermint, and anise, saffron, cocoa, nuts and seeds, artichoke, onion, spinach, and olives | • *Akkermansia*<br>• *Bacteroidota*<br>• *Blautia*<br>• *Roseburia* | Yes | • Prevention of enteropathogen growth<br>• Antioxidant properties reduce ROS levels<br>• Improved SCFA production | Varied dependent on compound | [70,71] |

1.3.6. Anti-Inflammatory and Immunomodulatory Effects of Prebiotic Consumption

At the crux of immunomodulation are the interactions of probiotic bacterial strains with the host gut epithelium and underlying immune cells. In the human gut, antigen-presenting immune cells such as dendritic cells sample the gut lumen for antigens with long filamentous processes. On these processes are Toll-like receptors and pattern recognition receptors that detect the microbe associated molecular patterns of different bacterial strains. Microbial adhesion to the colon epithelium facilitates these immune interactions and facilitates bacterial invasion of the gut epithelium. When gut pathogens invade the colonic epithelium a series of immune responses may occur. Toll-like receptors and nod-like receptors bind to pathogen-associated molecular patterns (molecules unique to pathogens) which can trigger intracellular signaling and production of inflammatory cytokines and promote the release of DNA-damaging reactive oxygen species. Antigen-presenting cells such as dendritic cells may activate T cells and B cells which have the potential to induce prolonged inflammation, especially if invasive pathogens are persistent in the gut lumen. It is thought that this chronic inflammation can induce CRC [72].

Studies have found that HMOs prebiotics can act directly as inhibitors to microbial pathogen adherence through their structural similarity to epithelial cell glycoproteins. HMOs have been shown to inhibit *Campylobacter*, *Vibrio cholera*, *Shigella*, *Salmonella*, *E. coli* toxins, and caliciviruses from binding to host cells in vitro [73]. HMOs also restrict the host dendritic cell response to antigens such as lipopolysaccharides by decreasing expression of TLR-4 and microbial pattern recognition receptors to maintain tolerance to gut microbes and avoiding destructive inflammatory responses [74].

Other prebiotics exert their anti-inflammatory activity via conversion to SCFAs by commensal probiotic strains. Propionic acid and butyric acid SCFAs inhibit chemokine release by host cells and the adhesion of several pathogens to the gut epithelium. They also reduce the formation and release of NO, and IL-6 and TNF-alpha inflammatory cytokines, while increasing the release of IL-10 and IL-8 anti-inflammatory cytokine [75]. SCFAs in the gut lumen bind FFA2 and HCA2 receptors (G protein-coupled receptors) of dendritic cells to initiate immunomodulation. Activation of downstream pathways stimulates activity of transcription factor NF-κB which can promote the expression of anti-inflammatory cytokines. Dendritic cells subsequently release IL-10 which stimulates the proliferation and differentiation of Treg cells. Dendritic cells also release IL-8 which functions with IL-10 to resolve inflammatory states. B cells are also stimulated by dendritic cells to release IgA antibodies which are secreted into the gut lumen and function to eliminate pathogens. It is important to note that the exact mechanism of B cell stimulation by dendritic cells to release IgA is still debated, and the precise changes to the host microbiome by increased IgA secretion are still unclear [76]. These various mechanisms are beginning to uncover the vastly complex interplay between the host immune system and microbiome. Further research is needed to determine the capacity for immunomodulation of different microbes in vitro and in vivo, and how the consumption of specific prebiotics alters these interactions.

## 2. Discussion: Studies for Prebiotics and Colorectal Cancer

To understand the relationship between prebiotics and colorectal cancer prevention, it is important to keep in mind that the effects of prebiotics are pleiotropic in the body. In the following study, the authors examined how metabolites of probiotic bacteria affect the growth of colon cancer cells in vitro. These metabolites, exopolysaccharides (EPSs), are secreted by many strains of bacteria and have been shown to behave as antioxidants with implications for cancer treatment [77]. Tukenemz et al. studied the anti-tumor effects of exopolysaccharides (EPSs) produced by probiotic bacteria on HT-29 colon cancer cells in vitro. Four different strains were used to produce and isolate EPS. Strains were isolated from infant feces, including *L. plantarum* GDR, *L. brevis* LB63 and *L. rhamnosus* E9: the other from yogurt, *L. delbrueckii ssp. Bulgaricus* B3 [35]. The results of the study indicated that EPS had anti-proliferative effects on the HT-29 cells in a time-dependent manner, with 48 h and 72 h having a greater effect than after 24 h. In addition, the study focused on

genes associated with the regulation of cell death, including Bax, Bcl-2, Caspase 3, Caspase 9, and Survivin. A significant decrease in the Bcl-2 gene and an increase in the Bax gene expression was observed as well. This data suggest that anti-apoptotic proteins Bcl-2 were unable to bind to pro-apoptotic protein Bax, and thus were unsuccessful in inhibiting its transfer to the mitochondrial membrane to cause cell death. Additionally, Caspase 3 and Caspase 9 gene expression was significantly increased, while Survivin expression, which acts to inhibit these genes via the p-53-survivin pathway, was decreased. Interestingly, the researchers noted that although EPS produced by all four strains had the capabilities to induce apoptosis, EPS containing the highest amount of mannose in sugar composition and low glucose content showed greater effects on induction of apoptosis in HT-29 cells [35]. Although these anti-tumor effects are intriguing and may seem promising, they were seen in vitro on isolated cells. The gut microflora is extremely diverse, varying from person to person, and the interaction of bacteria with prebiotics does not guarantee similar results during in vivo studies that involve humans [78]. Further studies should be conducted to determine if the consumption of certain prebiotic fibers enhances the populations of these EPS-producing strains of lactobacilli in CRC patients. If such a study is paired with tumor resection, molecular studies could also be performed to assess for apoptotic markers.

In a study by Qamar et al., rats were treated with 1,2-dimethylhydrazine dihydrochloride (DMH) or azoxymethane to induce aberrant crypt foci (ACF). These crypts form before colorectal polyps and are one of the earliest changes in the colon which may lead to CRC [79]. Animal groups which were treated with galacto-oligosaccharides (GalOS) produced by *Limosilactobacillus reuteri* showed resistance against body weight gain, which was induced by DMH in the control group. Prebiotic treatment groups also showed dose-dependent hindering in the manifestation of ACF in all parts of the colon. This is significant because it demonstrates that CRC can be prevented or slowed in its early stages with use of prebiotics. In addition, short-chain fatty acids (SCFAs), which are the byproducts of bacteria in the presence of prebiotics and play a role in intestinal homeostasis, were seen in higher concentrations in rats treated with prebiotics. The researchers also observed changes in harmful bacterial enzymes in the cecal and fecal samples. Lower contents of nitroreductase, ß-glucoronidase, and azoreductase enzymes, which can influence carcinogenesis progression, were observed in the GalOS and inulin-treated groups [79]. It has been noted in an investigation on rats' diets that these enzymes are increased with a high beef diet [80]. Intriguingly, there was also an observed change in bacterial populations. While the microbial concentrations of beneficial bacteria such as *bifidobacteria* and *lactobacilli* increased, the concentrations of harmful bacteria such as clostridia were observed to decrease in the GalOS and inulin-treated groups [81]. This once again demonstrates the complex nature of the diversity of the intestinal bacteria and the cascade of effects prebiotics can have on it.

Similarly, to the previously discussed study, Fernández et al. used an animal model and chemically induced tumors in rats using azomethane (AOM) and dextran sodium sulfate (DSS). However, these animals were prophylactically given 10% (*w/w*) GalOS-Lu, derived from lactulose, continuously in their drinking water prior to the chemical induction of tumors. Once the animals were sacrificed, the rats in the GalOS-Lu cohort showed a decrease in overall body weight compared to subjects in the control group that received normal diets plus DMH. Conversely, the researchers observed an increase in cecum weight which they attributed to stimulation of bacterial populations and the fermentative process in presence of the prebiotics. Similar to the study conducted by Qamar et al., the researchers observed a shift in the rats' intestinal microbiota. The effects of GalOS-Lu resulted in an increase in *Bacteroides* and a decrease in *Firmicutes* (note that the phylum *Bacteroidetes* has since been renamed to *Bacteroidota* and the phylum *Firmicutes* to *Bacillota).* Similar *Bacteroides/Firmicutes* ratios have been previously linked with obesity and type 2 diabetes mellitus [82]. Consequently, the increase in *Bacteroidoes* and decrease in *Firmicutes* has been used as a metabolic marker to identify healthy individuals. Fernández et al. also observed a statistically significant difference in the number of polyps between the control cohort and the rats in the GalOS-Lu cohort. Not only did the GalOS-Lu study group show a 57.5%

reduction in the number of polyps, there was also a 50.4% reduction in the mean affected tumor area [82]. Although this finding is promising, it is important to consider that not all polyps will convert to tumors and become malignant [33]. However, in common practice, doctors choose to remove all polyps to avoid the potential risk, so these findings may minimize future procedures for patients.

Effects of prebiotics examined were replicated in a study conducted by Lin et al. using a combination of germinated brown rice (GBR) and *Lactobacillus acidophilus, Bifidobacterium animalis* subsp. *lactis*, or both, on rats treated with DMH and DSS. Although there was no significant difference in body weight gain, which was seen in previous studies, the study did show that consumption of GBR alone, or in combination with probiotics, resulted in an increase protein expression of pro-apoptotic p53 and Bax, as well as a change in the Bax/Bcl-2 ratio. While Bcl-2 expression decreased, pro-apoptotic Caspase 3 expression was increased, with the most significant change seen in the group receiving the synbiotic combination of GBR and *L. acidophilus*. The cohort given GBR and *L. acidophilus* was also observed to have the most significant inhibition on the formation of ACF. This duplicated result supports the notion that prebiotics, when combined with probiotics, may inhibit early formation of colorectal cancer and may play a key role in its prevention. This inhibitory effect on ACF formation was not observed in rats fed a combination of GBR and *B. animalis* subsp. *lactis* which further confirms that not only the quantity, but the quality and the product selection also matter for prebiotics to exert the greatest efficacy. Previously, it was mentioned that not all polyps become malignant; similarly, not all ACF will convert to tumors. The extent of ACF malignancy is classified based on mucin, a substance that exists on the surface of the colon and protects the epithelium against mechanical or chemical damage. Alteration to the molecular barrier can be seen in patients with CRC, resulting in the increase in the tumor promoting sialomucins (SiMs), and an decrease in the tumor suppressing sulfomucins (SuMs) [83]. This balance in the mucins was also shown to be important in a study by Milosevic et al. looking at the expression of leptin expression. Here, it was shown that alteration to the SiMs or SuMs can affect the development of colorectal cancer; this could be an area to examine further in conjunction with prebiotics in the future [84]. In the study conducted by Lin et al., all cohorts being treated with a combination of GBR and prebiotics showed a significantly reduced number of SiM-ACF, while GBR alone reduced the number of mucin-depleted foci (MDF) [83]. This supports the concept that prebiotics can be used alone, or in combination with probiotics, in order to prevent early development of carcinogenesis. Another investigation also found that the treatment with DMH decreased the activity of superoxide dismutase (SOD), an antioxidant enzyme responsible for the defense against oxidative stress and reactive oxygen species in colon cells. All cohorts treated with GBR and probiotics showed a significant elevation in levels of SOD [85].

Several clinical trials (included in Table 2) have demonstrated similar results as those seen in rats; more studies are being conducted looking at prebiotics and synbiotics in adjunct with normal cancer treatments or for CRC prevention (shown in Table 3).

**Table 2.** Completed preclinical and clinical trials on prebiotics and cancer summary table.

| Study | Product | Cell Type | Inclusion/Exclusion Criteria | Advantages/or Limitations | Results |
|---|---|---|---|---|---|
| Tukenmez et al. (2019) [35] | Exopolysaccharides (EPSs) produced by four probiotic strains: *L. rhamnosus L. plantarum, L. brevis, and L. delbreueckii bulgaricus* | HT-29 colon cancer cells | - Not applicable | - Study performed on isolated colon cancer cells in vitro and may not have clinical significance but is thought to represent a suitable infected patient | - Time-dependent inhibition of proliferation of colon cancer cells via apoptosis<br>- Increased expression of Bax, Caspase 3, and Caspase 9<br>- Decreased expression of Bcl-2 and Survivin |
| Qamar et al. (2017) [79] | Galacto-oligosaccharides (GalOS) produced by *Limosilactobacillus reuteri* Inulin | Rats | - Not applicable | - Aberrant crypt foci are a good biomarker for detecting early onset of colorectal cancer<br>- Use of "true" prebiotics | - Dose-dependent protective effects against CRC<br>- Increase in body weight<br>- Decreased occurrence of aberrant crypt foci<br>- Increase in short-chain fatty acid production<br>- Lower contents of potential harmful enzymes such as nitroreductase, ß-glucoronidase, and azoreductase<br>- Increase in microbial population of Bifidobacteria and Lactobacilli<br>- Decrease in concentrations of Clostridia |
| Fernández et al. (2018) [82] | Galacto-oligosaccharides (GalOS) derived from lactulose (Lu) | Rats | - Not applicable | - Prophylactic use of prebiotics in water supply does not guarantee that all rats ingested identical amounts of treatment | - Decrease in body weight<br>- Increase in caecum weight due to stimulation of bacterial populations in presence of prebiotic<br>- Decrease in number of polyps and mean area affected<br>- Increase in short-chain fatty acid production<br>- Increase in Bacteroides<br>- Decrease in Firmicutes<br>- Decrease in pro-inflammatory bacteria |

**Table 2.** *Cont.*

| Study | Product | Cell Type | Inclusion/Exclusion Criteria | Advantages/or Limitations | Results |
|---|---|---|---|---|---|
| Lin et al. (2018) [83] | Germinated brown rice (GBR) combined with *Lactobacillus acidophilus* and *Bifidobacterium lactis* | Rats | - Not applicable | - Germinated brown rice is easily attainable to the common consumer | - No significant change in body weight<br>- Increased protein expression of pro-apoptotic p53 and Bax<br>- Decreased Bcl-2 and Caspase 3 expression<br>- Decreased number of aberrant crypt foci and sialomucin<br>- Increased levels of superoxide dismutase |
| Ohara et al. (2018) [86] | Fructo-oligosaccharides and *Bifidobacterium longus* | Humans | - Criteria was not specified. However, researchers mention that healthy adults participated in the study with a mean age of 60.2 years. The colon cancer cells were cultured in SCFA to monitor the growth inhibition. | - Healthy adults and cultured colon cancer cells<br>- Source of product was yogurt, which is easily attainable to the common consumer | - Increase in total amount of short-chain fatty acids<br>- Suppression in Bacteroides fragilis enterotoxin (ETBF)<br>- Increase in Bifidobacteria |
| Theodoropoulos et al. (2016) [87] | Synbiotic Forte: *Pediococcus pentoseceus* 5–33:3, *Leuconostoc mesenteroides* 32–77:1, *L. paracasei* ssp paracasei 19 and *L. plantarum* 2362, as well as 2.5 g inulin, oat bran, pectin and resistant starch | Humans | - Inclusion: Both genders between ages 18 and 80 years. Those with histological documentation of cancer of the colon or rectum, operated between July 2008 and April 2012.<br>- Exclusion: pregnant participants, patients with hereditary cancer, history of inflammatory bowel disease, metastatic disease, requirement of permanent or temporary stoma, and those not able to tolerate liquid diet by the 5th postoperative day. | - Double-blinded, prospective, randomized control trial<br>- Study included different types of colorectal resections, which can be applicable to a larger population<br>- Questionnaires are often subjective and may distort data due to bias | - Early post-operative synbiotic administration led to improved Gastro-Intestinal Quality of Life Index (GIQLI) scores<br>- Significant improvement in diarrhea in patients having undergone colectomy for colorectal cancer |

**Table 2.** *Cont.*

| Study | Product | Cell Type | Inclusion/Exclusion Criteria | Advantages/or Limitations | Results |
|---|---|---|---|---|---|
| Molan et al. (2014) [88] | First Leaf (FL) (blackcurrant extract, lactoferrin, lutein) | Humans | - Inclusion: 20–60 years; healthy volunteers.<br>- Exclusion: history of gastrointestinal disease other than appendicitis, alcohol consumption (more than two units/day), smoking, endocrine disease, cancer, vascular disease, use of antibiotics or laxatives within previous month, recent surgery, and medication use. | - Participants were instructed to keep usual dietary habits and to avoid use of vitamins, anthocyanins (avoid red fruits or vegetables, berries, grapes, red wine, and berry juices), and fermented dairy products | - Decreased fecal pH<br>- Increase in in Lactobacillus and *Bifidobacterium* spp. populations<br>- Significant reduction in fecal Clostridium and *Bacteroides* spp. populations<br>- Decreased activity of beta-glucuronidase in feces<br>- Increased activity of beta-glucosidase in feces |
| Costabile et al. (2012) [89] | Polydextrose (PDX) | Humans | - Inclusion: 18–50 years old, BMI 19–25 kg/m2, good general health.<br>- Exclusion: evidence of physical or mental disease, major surgery, history of substance abuse, severe allergy, history of abnormal drug reaction, smoking, pregnant, or lactating. | - Crossover study in European population<br>- PDX was not completely fermented by colonic bacteria<br>- No change in fecal butyrate levels was observed<br>- PDX did not decrease quality of life (as measured by survey response); did decrease abdominal discomfort<br>- Decline in fecal water genotoxicity | - PDX supplementation decreased levels of C. histolyticum; increased levels of R. intestinalis and C. leptum<br>- PDX treatment led to significantly less fecal water-induced genotoxic damage to HT29 DNA |

**Table 2.** *Cont.*

| Study | Product | Cell Type | Inclusion/Exclusion Criteria | Advantages/or Limitations | Results |
|---|---|---|---|---|---|
| Lanza et al. (2007) [90] | Dietary (low-fat, high-fiber, high-fruit, high-vegetable) | Humans | - Inclusion: 35 years or older, at least one histologically confirmed large-bowel adenomatous polyp removed during colonoscopy within previous 6 months.<br>- Exclusion: history of colorectal cancer, surgical resection of adenomas, bowel resection, polyposis syndrome, IBD, >150% recommended body weight, take lipid-lowering drugs, and medical or dietary restrictions that would limit ability to complete study. | - 1905 subjects completed study<br>- Reduction in fat intake to ≤20% of total calories<br>- 20% reduction in consumption of red and processed meats<br>- 18 g dietary fiber per 1000 kcal<br>- 3.5 servings of fruits and vegetables per 1000 kcal | - No significant difference in adenoma recurrence<br>- Potential confounders: trial duration, timing in life course, end point, and nature of intervention |
| Limburg et al. [91] | Prebiotics arm: oligofructose-enriched inulin taken 6 g 2×/day (Beneo Synergy 1) | Humans | - Inclusion: Adults 40 years and older with increased risk for CRC and the presence of ≥1 aberrant cryptic foci (see reference for full criteria) | - The study's primary and secondary outcome measures incorporate both endoscopic and molecular diagnostics<br>- Participants were randomized with regard to history of CRC and current presence of aberrant cryptic foci<br>- Randomized, controlled, and double-blinded study<br>- Small cohort sizes of ~20 may skew results | - Significant reduction in aberrant cryptic foci in the inulin prebiotic arm |

**Table 3.** Current ongoing or unpublished clinical trials on prebiotics and colorectal cancer.

| Title of Ongoing Study | Study Type | Interventions | Subjects | Primary Location | Estimated Study Completion Date | ID Number |
|---|---|---|---|---|---|---|
| Prebiotics and Probiotics During Definitive Treatment with Chemotherapy-radiotherapy SCC of the Anal Canal (BISQUIT) [92] | Phase II randomized trial Double blinded | Chemotherapy with or without prebiotics and probiotics | Patients with localized anal squamous cell cancer, 75 participants | Sau Paulo, SP, Brazil | February 2024 | NCT03870607 |
| Fiber to Reduce Colon Cancer in Alaska Native People [93] | Randomized trial Quadruple blinded | 70 g high amylose maize starch versus 70 g of amylopectin corn starch | Alaskan natives, 40–65 years old, 60 participants | Anchorage, Alaska, United States | January 2023 | NCT03028831 |
| Prebiotic Effect of Eicosapentaenoic Acid Treatment for Colorectal Cancer Liver Metastases [94] | Observational prospective cohort | 4 g daily of pure EPA-EE as soft gel capsules | Patients with colon cancer liver metastasis, 250 participants | Leeds, United Kingdom | July 2025 | NCT04682665 |
| Effect of the Use of Symbiotics in Patients With Colon Cancer [95] | Phase IV randomized trial Quadruple blinded | 6 g of synbiotics supplemented twice daily | Patients over 18 years of age, with colorectal and head and neck cancer who will undergo a tumor resection | Belo Horizonte, MG, Brazil | December 2022 | NCT04874883 |
| Fiber-rich Foods to Treat Obesity and Prevent Colon Cancer [96] | Randomized trial Double blinded (Investigator and Outcomes Assessor) | 250 g of legumes 2x/day for 3 months followed by 250 g of legumes 1x/day for an additional 3 months | Free-living adults 40–75 years old, BMI 25–40, colonoscopy within 3 years that found ≥1 adenoma >0.5 cm | Emory University | 31 December 2024 | NCT04780477 |

The currently ongoing clinical studies (Table 3) each have unique goals and endpoints. One study (NCT03870607) is aimed to measure the response rates of patients with squamous cell carcinoma of the anal canal; the comparison is treatment with standard of care (chemotherapy + radiation therapy) versus standard of care with daily prebiotic and probiotic consumption [92]. Another ongoing study is looking at the prevention of colon cancer by giving patients either fully digestible corn starch or resistant corn starch as daily supplements (NCT03028831). The primary outcome measure is proliferation of colonic epithelium (a biomarker for CRC risk) [93]. A third study is observing the effects of eicosapentaenoic acid (an omega-3 fatty acid commonly found in fish oil) on microbiome composition, stool composition, immune cell prevalence and immune checkpoint regulators in metastatic tissue, among others (NCT04682665) [94]. An additional ongoing study is measuring the effects of synbiotic treatment on the frequency of osmotic diarrhea in patients being treated for CRC (NCT04874883) [95]. The final open study (NCT04780477) mentioned in Table 3 is examining the effect of a high-fiber controlled diet in a population of patients who previously developed adenoma of the colon and have a BMI of 25–40 kg/m$^2$ [96]. One arm is receiving the controlled diet while an additional arm receives the control diet with legumes as a fiber source. At this time, there have not been updates nor results posted from these studies.

In a research study conducted by Ohara et al., human feces were sampled and analyzed before and after a diet of yogurt containing *Bifidobacterium longum* alone and in combination with a prebiotic, fructo-oligosaccharide. This study showed that the intake of both prebiotic and probiotic aided in preventing colorectal cancer [86]. The findings were consistent with the data seen in the isolated cancer cells and rats with both groups seeing a significant increase in the total amount of SCFAs [86]. In addition, the detection rate of *Bacteroides fragilis* enterotoxin (ETBF) and growth of putrefactive bacteria was suppressed. Interestingly, the *Bifidobacterium* detection rate was noted to be higher in the synbiotic group than in the probiotic only group. This can be explained by the synergistic mechanism in which prebiotics stimulate the growth of beneficial gut bacteria [86,97]. In a separate study by Theodoropoulos et al., the effects of prebiotics were also tested on patients who had undergone a colectomy procedure due to cancer. The aim of the study was to assess the participants' Gastro-Intestinal Quality of Life Index (GIQLI) at 1, 3, and 6 months postoperatively, and to assess whether synbiotics could improve functional bowel symptoms such as diarrhea and constipation using the European Organization for Research and Treatment of Cancer (EORTC) 30-item questionnaire (QLQ-C30). Participants were supplemented with synbiotics through the Synbiotic Forte formulation which included the following: *Pediococcus pentoseceus* 5–33:3, *Leuconostoc mesenteroides* 32–77:1, *Lacticaseibacillus paracasei* ssp paracasei 19 and *Lactiplantibacillus plantarum* 2362, as well as 2.5 g inulin, oat bran, pectin and resistant starch. The results indicated that the cohort receiving synbiotics had a better GIQLI score compared to those receiving placebo. Although there was no significant effect on constipation, the EORTC QLQ-C30 questionnaire demonstrated that administration of synbiotics in patients who had undergone a colectomy for colorectal cancer led to a significant improvement in diarrhea [87]. This may suggest that use of synbiotics may be helpful to alleviate symptoms of irritable bowel syndrome (IBS), which has similar symptoms of cramping, bloating, constipation, and diarrhea, as seen in CRC [98,99]. These findings indicate that prebiotics may not only help with the prevention of CRC but may also be beneficial for individuals with certain risk factors for disease and CRC patients facing post-operative GI dysfunction.

## 3. Conclusions

Prebiotics, when administered alone or in combination with suitable probiotics as synbiotics, have the potential to prevent tumor formation, which has been demonstrated in animal and human studies. The anti-proliferative effects of prebiotics have been demonstrated in vivo in rodents by the inhibition of formation of aberrant crypt foci and colonic polyps, as well as changes in gene expression such as an increase in pro-apoptotic Bax, Cas-

pase 3, and Caspase 9, and the decrease in expression of anti-apoptotic Bcl-2 and Survivin in vitro in cells. Prebiotics have also shown to alter bacterial concentrations in both animal in vivo and human in vitro testing, raising the levels of beneficial bacteria such as *Lactobacillus*, *Lacticaseibacillus*, *Lactiplantibacillus*, *Levilactobacillus*, *Limosilactobacillus Bifidobacterium*, and *Bacteroides* while lowering the levels of harmful bacteria such as *Clostridium* spp., as well as harmful enzymes such as nitroreductase, ß-glucoronidase, azoreductase. Prebiotics have shown to not only be beneficial in preventing the development of colorectal cancer but have also been demonstrated to be favorable in improving the symptoms of those who have been affected by the disease. Overall, prebiotics can improve gastrointestinal health when used correctly. However, as the scientific community continues to explore the benefits of prebiotics, challenges remain for the public consumer regarding the regulation and selection of products. In the practice of medicine, patients may ask questions regarding prebiotics, probiotics, or synbiotics, and it is important to decide the correct dose, duration and product that benefits the patient. It is crucial to continue to research and develop the optimal use of prebiotics as a tool in the battle against colorectal cancer.

**Author Contributions:** Conceptualization, R.R.D.; Writing—A.S., M.M., M.D., D.L.A. and R.R.D. Original draft preparation, A.S., M.M., M.D., D.L.A. and R.R.D.; Writing—review and editing, R.R.D., P.L., D.L.A. and B.S.; Visualization, A.S., M.M. and M.D.; Supervision, R.R.D. and P.L.; Project administration, R.R.D. and P.L. All authors have read and agreed to the published version of the manuscript.

**Funding:** This research received no external funding.

**Institutional Review Board Statement:** Not Applicable.

**Informed Consent Statement:** Not Applicable.

**Data Availability Statement:** Not Applicable.

**Conflicts of Interest:** The authors declare no conflict of interest.

**Literature Search:** The literature cited in this paper was found primarily through PubMed and ClinicalTrials.gov searches. Several works on novel prebiotics could not be reached on these sites alone and further searches were performed on Google Scholar from the year 2018–present to discover publications with stronger emphasis on molecular engineering and biotechnology. Older literature going back to 1995 was required to provide complete background information.

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
