# Peer review of "A Current Review on the Role of Prebiotics in Colorectal Cancer"

_biologics, doi:10.3390/biologics3030012_

Round 1

Reviewer 1 Report (Previous Reviewer 3)

The review is well written, very timely and comprehensive especially with the new additions. My compliments to the authors. It should be published in its present form. 

Two minor remarks: Line 391: the correct name is Salmonella enterica serv. Typhimurium or short Salmonella Typhimurium. Line 531/532: one expression should be deleted.

Author Response

Reviewer 1 comments

The review is well written, very timely and comprehensive especially with the new additions. My compliments to the authors. It should be published in its present form.

Thank you to all reviewers for taking the time to examine our manuscript and give feedback.

Reviewer 2 Report (Previous Reviewer 1)

The authors have addressed my concerns, and I have no further comments. 

Author Response

Reviewer 2 comments

The authors have addressed my concerns, and I have no further comments.

Thank you to all reviewers for taking the time to examine our manuscript and give feedback.

Reviewer 3 Report (New Reviewer)

The work is complete, understandable and quite interesting. The issue addressed is very current and you try to bring out how the studies conducted in the literature can elucidate the connection between beneficial effects for the host and the administration of probiotics and prebiotics.

I list below the small corrections and clarifications that in my opinion the authors should make to the manuscript:

- add a paragraph or an explanation in an existing paragraph of how the search for the articles was conducted to write the review (if you use pubmed or other, years considered, other parameters...)

-       Move lines 80-83 to lines 33-45, where you are describing the definitions

-       Lines 115-124: add the difference of synbiotics: synergistic of functional with a brief explanation (ISAPP)

-       Line 130: remove ..

-       Line 131: write in the correct way Caco-2

-       Line 140: write in the proper way CO2 (CO2)

-       Line 164: “Other tests include blood test and fecal samples”. Maybe you can add some example of the monitored parameters

-       Lines 234-236: the degree of polymerization of the molecules could be added as important information for categorize the molecules as inulin, fos…

-       Line 298: insert the dot

-       Line 322, 328, 475, table 2, 583-584: use the new nomenclature of Lactobacilli and write correctly their names.

-       Line 501: maybe you can indicate the new nomenclature for Firmicutes and Bacteroidetes

-       Line 604: its not correct to say “lowering the levels of harmful bacteria such as Firmicutes”, because not all the firmicutes (now Bacillota) are harmful, i.e. Lactobacillus strains. I suggest to write the sentence in the proper way

English is very good and fluent, understandable

Author Response

Reviewer 3 comments

The work is complete, understandable and quite interesting. The issue addressed is very current and you try to bring out how the studies conducted in the literature can elucidate the connection between beneficial effects for the host and the administration of probiotics and prebiotics.

I list below the small corrections and clarifications that in my opinion the authors should make to the manuscript:

- add a paragraph or an explanation in an existing paragraph of how the search for the articles was conducted to write the review (if you use pubmed or other, years considered, other parameters...)

Response: Thank you. This was placed in a separate paragraph before the references. Please see the added text regarding the literature search in lines 638-642.

-       Move lines 80-83 to lines 33-45, where you are describing the definitions

Response: Thank you for this suggestion. We have adjusted the text in this section to achieve improved organization and clarity. Please see the red text in lines 48-58 and 89-90.

-       Lines 115-124: add the difference of synbiotics: synergistic of functional with a brief explanation (ISAPP)

Response: Thank you for this comment. Please see the updated text in lines 121-128 which includes the updated definitions of synbiotics.

-       Line 130: remove ..

Response: Thank you for pointing out this error. It has been corrected.

-       Line 131: write in the correct way Caco-2

Response: Thank you, this correction has been made.

-       Line 140: write in the proper way CO2 (CO2)

Response: Thank you, this correction has been made.

-       Line 164: “Other tests include blood test and fecal samples”. Maybe you can add some example of the monitored parameters

Response: Thank you for this comment. We added details that will help readers better understand the screening tools available. Please see lines 165-171.

-       Lines 234-236: the degree of polymerization of the molecules could be added as important information for categorize the molecules as inulin, fos…

Response: Thank you, this information has been included in lines 240-242.

-       Line 298: insert the dot

Response: Thank you this has been corrected.

-       Line 322, 328, 475, table 2, 583-584: use the new nomenclature of Lactobacilli and write correctly their names.

Response: Thank you, these names have been updated where applicable.

-       Line 501: maybe you can indicate the new nomenclature for Firmicutes and Bacteroidetes

Response: Thank you for the good suggestion. This has been noted in lines 514-515 but not edited throughout the remaining text to maintain clarity.

-       Line 604: its not correct to say “lowering the levels of harmful bacteria such as Firmicutes”, because not all the firmicutes (now Bacillota) are harmful, i.e. Lactobacillus strains. I suggest to write the sentence in the proper way.

Response: Thank you, “Firmicutes” was removed to correct this.

This manuscript is a resubmission of an earlier submission. The following is a list of the peer review reports and author responses from that submission.

Round 1

Reviewer 1 Report

The review article on the role of Prebiotics in Colorectal cancer by Shrifteylik et al is an interesting read and summarizes the studies that provide the effects of prebiotics on human health w.r.t. CRC. Practically, I have no comments, however, there are few suggestions that I think can be beneficial for the potential readers.

1. Summarize the classification of prebiotics as a table or a figure showing their different types. e.g. Glycan-based, non-glycan-based, etc.

2. A general overview highlighting the mechanism of how prebiotics works with a special focus on CRC.

3. Although the authors have highlighted the potential side effects of prebiotics on page 4, the text is buried deep inside section 1.1 and barely noticeable. In my opinion, the limitations and side effects of prebiotics deserve a separate section. Moreover, the side effects highlighted in the current manuscript are too general, therefore, authors should look into the reported studies to check if there are any additional side effects.

4. Is there any correlation between the CRC stage and prebiotics?

5. Finally, the authors should also mention if the dose can regulate the positive or negative effects of the prebiotics.

Page 7, Line 278: “….weight, gain,….” → “…..weight gain,…..”

Reviewer 2 Report

I would like to thank MDPI to offer me the opportunity of reviewing the manuscript entitled “Feed Your Gut, Save Your Butt – A Review of the Role of  Prebiotics in Colorectal Cancer”. The review is interesting but I would like to point out some flaws that should be revised by authors.

First, I would deeply recommend the authors to change the title, mostly by the reason they are exploring the benefits of prebiotics upon colorectal cancer, which is not a joke. But also because in my opinion the use of this kind of language in a scientific paper is not appropriated at all.

The current review is really similar to one recently published in Microorganisms (https://www.mdpi.com/2076-2607/9/6/1325) and does not brings novelty to the current knowledge regarding prebiotics and colorectal cancer. 

Moreover, I would kindly suggest the authors in order to publish the manuscript to arrange better the structure of the text, use newer and better references. 

Reviewer 3 Report

This is a comprehensive and timely review on the effects of prebiotics on colon cancer. It includes description how the term was defined and that the advancement of data requires constant evolution of the definition. The authors describe some in vitro experiments for which the significance for the in vivo situation is not clear, however, at the end of the chapter they point this clearly out. Thus, the authors handle their subject very critical which helps the reader to judge the present status of research. I have some minor points that the authors might address:

1: In the second sentence of the summary the authors write about association of intestinal bacteria with development of cancer. However, the entire review concentrates on prevention of cancer. The authors might want to remove this sentence as it is a bit misleading.

2. I have the impression that most of the effects of prebiotics are due to the conversion of some of the fibres to SCFAs. It would be good to mention this also in the summary. In addition, the effects of SCFAs are extremely pleiotropic and only some of the effects are mentioned. It wood be good to mention this on line 63 already and describe more of such activities more extensively. Possibly, a small table would be appropriate.

Reviewer 4 Report

Title: Feed Your Gut, Save Your Butt – A Review of the Role of Prebiotics in Colorectal Cancer

Authors: Anna Shrifteylik, Morgan Maiolini, Daniel L. Austin, Bobban Subhadra, Purushottam Lamichhane, Rahul R. Deshmukh

Overview and general recommendation:

Presented manuscript aims to discuss the definition and scope of prebiotics by reviewing the studies that provide insight to their effects on the human health in the context of colorectal cancer.

Although the subject of the manuscript is very important, as prebiotic are functional, por-healthy ingredients that might be used in creating functional product to prevent or postpone the colorectal cancer, the manuscript needs to be improved.

The review manuscript should present a subject in  comprehensive way. That is why I would recommend adding also the information about the role of SCFAs, antiflammatory and immunomodulatory effects of prebiotics, the mechanisms of the increment of  the production of mucus and IgA in the mucus, mucosal hyperplasia…

Below I give also some concerns that require review:

Mainor comments:

Title – “Feed Your Gut, Save Your Butt – A Review of the Role of Prebiotics in Colorectal Cancer” – the title however is concise, specific and relevant, but in my opinion is not suitable for scientific journal, as Biologics is, I would recommend living only the second part.

Affiliations – authors should give a full University or Company names including City and country, as well as give details about the Corresponding author not “Communicating authors”, and the authors contribution is given not at the beginning of the article, but on the end – please fill the suitable place in the template.

Line 40 – “industry” – different font.

Line 62 – “These molecules are used are commonly used for energy for cell processes” – sentence need to be rewritten.

Line 64 – why in this place and why only HMOs are described?

Chapter: Introduction: Definition of Prebiotics and Its Evolution - there is lack of the conclusion abut what are the prebiotics now, are there class of prebiotics, what ingredients might be classified as prebiotics?  The newest prebiotics for example are the mixture of nondigestible polysaccharides and protein. The review article in order to be useful needs to provide up-to-date information.

Line 106 – “According to the National Health Interview Survey (NHIS), the use of prebiotics or probiotics was four times higher in the year of 2012 than it was in 2007…” – and what is a recent situation - almost 10 years have passed since 2012?

Lines 119-120 – “Lastly, they influence possible pathogenic bacteria; since some of the gut microbes can ‘eat’ the pathogens…” – please replace the word "eat" with more appropriate for scientific articles.

Line 123 “….allow them to flourish ” - please replace the word "flourish" with more appropriate for scientific articles.

Lines 142-155 – in my opinion the prebiotics examples are not described properly, please add into the description of inulin, GOS and lactulose information about a chemical composition, which explain why they are not digested in human system, and what is also the mechanism of their action.

Table 1 – “Common uses of the three recognized prebiotics” – please add information where those uses are? Are they drug or food?

Lines 167-168 – “This includes effects like cramping, nausea, flatulence, and bloating.” – please add an explanation?

Line 224, 297 - Some references are missed in the Reference list – ex. Lin et al., Fernández et al.

Conclusion - Microorganisms names should be written in italics